# Work-Related Psychosocial Factors and Global Cognitive Function: Are Telomere Length and Low-Grade Inflammation Potential Mediators of This Association?

**DOI:** 10.3390/ijerph20064929

**Published:** 2023-03-10

**Authors:** Caroline S. Duchaine, Chantal Brisson, Caroline Diorio, Denis Talbot, Elizabeth Maunsell, Pierre-Hugues Carmichael, Yves Giguère, Mahée Gilbert-Ouimet, Xavier Trudel, Ruth Ndjaboué, Michel Vézina, Alain Milot, Benoît Mâsse, Clermont E. Dionne, Danielle Laurin

**Affiliations:** 1Centre d’excellence sur le vieillissement de Québec (CEVQ), CIUSSS-Capitale Nationale, Québec, QC G1S 4L8, Canada; 2Faculty of Medicine, Université Laval, Québec, QC G1V 0A6, Canada; 3Centre de Recherche du CHU de Québec—Université Laval, Québec, QC G1S 4L8, Canada; 4VITAM, Centre de Recherche en santé Durable, Québec, QC G1S 4L8, Canada; 5Institut sur le Vieillissement et la Participation Sociale des Aînés, Université Laval, Québec, QC G1S 4L8, Canada; 6Canada Research Chair in Sex and Gender in Occupational Health, Université du Québec à Rimouski, Campus de Lévis, Lévis, QC G6V 0A6, Canada; 7School of Social Work, University of Sherbrooke, Sherbrooke, QC J1K 2R1, Canada; 8Institut National de Santé Publique du Québec (INSPQ), Québec, QC G1V 5B3, Canada; 9École de Santé Publique de l’Université de Montréal, Montréal, QC H3N 1X9, Canada; 10Faculty of Pharmacy, Université Laval, Québec, QC G1V 0A6, Canada

**Keywords:** psychosocial stressors at work, effort–reward imbalance, demand–control–support, inflammatory biomarkers, telomere, IL-6, CRP, cognitive function, mediation analysis

## Abstract

The identification of modifiable factors that could maintain cognitive function is a public health priority. It is thought that some work-related psychosocial factors help developing cognitive reserve through high intellectual complexity. However, they also have well-known adverse health effects and are considered to be chronic psychosocial stressors. Indeed, these stressors could increase low-grade inflammation and promote oxidative stress associated with accelerated telomere shortening. Both low-grade inflammation and shorter telomeres have been associated with a cognitive decline. This study aimed to evaluate the total, direct, and indirect effects of work-related psychosocial factors on global cognitive function overall and by sex, through telomere length and an inflammatory index. A random sample of 2219 participants followed over 17 years was included in this study, with blood samples and data with cognitive function drawn from a longitudinal study of 9188 white-collar workers (51% female). Work-related psychosocial factors were evaluated according to the Demand–Control–Support and the Effort–Reward Imbalance (ERI) models. Global cognitive function was evaluated with the validated Montreal Cognitive Assessment (MoCA). Telomere length and inflammatory biomarkers were measured using standardised protocols. The direct and indirect effects were estimated using a novel mediation analysis method developed for multiple correlated mediators. Associations were observed between passive work or low job control, and shorter telomeres among females, and between low social support at work, ERI or iso-strain, and a higher inflammatory index among males. An association was observed with higher cognitive performance for longer telomeres, but not for the inflammatory index. Passive work overall, and low reward were associated with lower cognitive performance in males; whereas, high psychological demand in both males and females and high job strain in females were associated with a higher cognitive performance. However, none of these associations were mediated by telomere length or the inflammatory index. This study suggests that some work-related psychosocial factors could be associated with shorter telomeres and low-grade inflammation, but these associations do not explain the relationship between work-related psychosocial factors and global cognitive function. A better understanding of the biological pathways, by which these factors affect cognitive function, could guide future preventive strategies to maintain cognitive function and promote healthy aging.

## 1. Introduction

Dementia is emerging as a public health priority due to the aging of the population [1]. In 2020, the Lancet Commission on Dementia Prevention, Intervention, and Care reiterated its position on the relevance of a life-course approach for the identification of modifiable risk factors, targeting those contributing favorably to cognitive reserve [2]. A cognitive reserve refers to the neurobiological maintenance and adaptability of the brain, enabling preservation of cognitive function in everyday activities, despite the presence of brain pathology [3]. Modifiable factors at midlife, such as occupational complexity, frequent social contact, and intellectual and social activities, have been suggested to benefit cognitive reserve [2,3].

The role of work-related psychosocial factors in affecting cognitive function has generated increasing interest in recent years. It has been suggested that these factors could help build cognitive reserve when work involves high intellectual complexity, decision making, creativity, skill discretion and development, and high social interaction with coworkers and supervisors. Systematic reviews and meta-analyses have reported associations between high complexity at work and better cognitive performance [4,5,6], and low risk of dementia [5,6]. However, work-related psychosocial factors that are perceived as stressful may have detrimental health effects [7,8,9]. Chronic psychosocial stressors at work could increase the synthesis of inflammatory biomarkers, including C-reactive protein (CRP) and interleukin-6 (IL-6), which, in turn, could induce low-grade inflammation [10]. This low-grade inflammation promotes the oxidative stress that is involved in the DNA degradation process and telomere shortening [11]. Telomeres are protective and stabilizing structures, made up of repeated non-coding DNA sequences, located at the end of chromosomes [12]. Telomere shortening is part of the natural cellular life cycle, but an increased rate of shortening is associated with premature cellular aging [11]. Both low-grade inflammation and shorter telomeres have been associated with a cognitive decline [13,14,15,16,17,18,19,20] and dementia [21,22,23,24] in longitudinal studies, but these findings have not consistently been replicated [25,26,27,28]. Untangling biological mechanisms by which work-related psychosocial factors affect cognitive function is important to distinguish those that have the potential to increase cognitive reserve from those that could precipitate cognitive decline.

Work-related psychosocial factors have been conceptualized by two recognised theoretical models. According to the Demand–Control–Support (DCS) model [29,30], exposure to high psychological demands, combined with low job control, constitutes high job strain, increasing physiological and psychological stress responses, whereas exposure to low psychological demands, combined with high job control, constitutes a low job strain [29]. Exposure to high psychological demands, combined with high job control, defines active work, which provides high developmental and intellectual opportunities, whereas exposure to low psychological demands, combined with low job control, defines passive work [29]. Exposure to low social support at work from colleagues and supervisors, the third component of this model, can amplify the adverse effect of high job strain on health [30]. Based on the Effort–Reward Imbalance (ERI) model [31], exposure to an imbalance between high efforts spent at work and low economic, social, or organisational rewards obtained in return, is associated with a stress condition that has detrimental health effects. Exposures to high job strain and to ERI have both been associated with an increased risk of poorer health outcomes, such as hypertension [32], cardiovascular diseases [33,34], diabetes [35], and mental health problems [7,8,9].

Longitudinal studies provide evidence linking passive work [36,37,38] and low job control [36,37,38,39,40] to worse global cognitive function. The association of high job strain with lower global cognitive function has been documented [36,39], although inconsistently [37,38,40]. Some longitudinal studies have reported associations between work-related psychosocial factors from the DCS or ERI models and higher concentrations of inflammatory biomarkers [41,42,43]. We previously reported an association between exposure to high job strain, combined with low social support or exposure to ERI, and an inflammatory index, especially among males [44]. To our knowledge, no previous longitudinal study has examined the association of these work-related psychosocial factors with telomere length. Additionally, the mediating effects of telomere length or inflammatory biomarkers have not been examined in previous studies, regarding the relationship between work-related psychosocial factors and cognitive function. Furthermore, sex differences have not been thoroughly examined, even though several sex differences have been reported with respect to exposure to work-related psychosocial factors [45], inflammatory biomarkers [46], telomere length [47], cognitive function [48], decline over time [49], and the prevalence of dementia [50].

This study aimed to evaluate the interplay between work-related psychosocial exposures, global cognitive function, telomere length and inflammatory biomarkers, overall and by sex. Specifically, this study examined the total, direct, and indirect effects of work-related psychosocial factors on global cognitive function through telomere length and an inflammatory index, combining CRP and IL-6 within the PROspective Quebec (PROQ) study on work and health [51].

## 2. Materials and Methods

### 2.1. Study Design and Data Collection

The PROQ study is a longitudinal occupational cohort study, designed to evaluate the effects of work-related psychosocial factors on cardiovascular and mental health outcomes. The study population at baseline included 9188 white-collar workers (initial participation: 75%; 51% females), recruited in 19 public and semi-public organizations from Quebec City, Canada, 1991–1993 (T1). Two follow-ups were carried out after 8 (T2, 1999–2000) and 24 years (T3, 2015–2018). Methodological details of the study have been described elsewhere [51]. In brief, at each measurement time point, data collection included a self-reported questionnaire, a face-to-face interview, and anthropometric and clinical evaluations. At T3, two new components were added to the main study: one on mental health and the other on biomarkers. Before recontacting participants for T3, one third of all participants at T1 were randomly selected for the study of biomarkers, with the exclusion of 207 participants who had either died between T1 and T2 (n = 117), did not want to be recontacted for the follow-up (n = 77) or were not traceable at T3 (n = 13) (Figure 1). When participants were recontacted at T3, most of them were retired (77%). To ensure sufficient statistical power for the planned analysis, a second random selection was therefore performed, including 50% of participants still working at T3. Finally, 3411 participants were selected for the study of biomarkers. Among those, 959 were not included in the present analysis, because they either died between T2 and T3 (n = 233), were lost to follow-up (n = 82), refused to participate in any part of the T3 data collection (n = 321), declined to participate in the face-to-face interview including the cognitive assessment but granted access to medical data only (n = 218), or completed the self-reported questionnaire only (n = 334). Finally, 4 participants were excluded because of missing data on cognitive function, leaving 2219 participants for the present analysis. All participants, or their respondents, provided written informed consent. The CHU de Québec-Université Laval Research Ethics Board reviewed and approved this study.

### 2.2. Work-Related Psychosocial Factors

For this study, we used work-related psychosocial factors measured at T2 for baseline exposure, because data from both models were available. Unlike the DCS model, the ERI model was not yet published at T1. Psychological demands (9 items), job control (9 items), and social support at work (11 items) were measured using the validated French adaptation of the Karasek questionnaire [29,30,52,53,54]. Reward was measured using 9 of the 11 items from the validated French adaptation of the Siegrist ERI questionnaire [31,55,56]. Each item was measured on a four-point Likert scale, and scores were calculated for each factor by summing the items. The scores for psychological demands and job control were dichotomised according to the median of a representative sample of Quebec workers [57], and four categories were created: high job strain (high demands and low control), low job strain (low demands and high control), active work (high demands and high control), and passive work (low demands and low control) [29]. To evaluate exposure to high job strain, the reference group comprised the three other categories, and the same procedure was used to evaluate the exposure to passive work. The score of social support at work was dichotomized at the median of the study sample at T2. Exposure to iso-strain was defined by the exposure to high job strain, combined with the exposure to lower-than-median social support at work. The ERI ratio was created by dividing the score of psychological demands, as a proxy of the Siegrist effort scale, by the score of reward. The psychometric qualities of this ERI version have been demonstrated [58]. Participants with a ratio greater than 1 were categorized as exposed to ERI.

### 2.3. Cognitive Function

Global cognitive function was evaluated with the validated Montreal Cognitive Assessment (MoCA) [59], during a standardised face-to-face interview at T3. The MoCA is a 30-point screening test for mild cognitive impairment or cognitive dysfunction, evaluating visuospatial abilities, executive function, short-term memory recall, attention, concentration, working memory, language and orientation in time and space. The sensitivity, specificity, and positive and negative predictive values of MoCA are about 90% compared to a clinical diagnosis of mild cognitive impairment made by a neurologist [59]. The screening performance for mild cognitive impairment of MoCA is superior to that of the Mini-Mental State Examination and other tests for global cognitive function, justifying its use in cognitively healthy adults [59,60,61].

### 2.4. Blood Samples

Non-fasting blood samples were taken at T3 by a trained nurse, following a standardised protocol. Venous blood samples were centrifuged within 24 to 72 h, and aliquots of the serum, plasma and buffy coats were kept frozen at minus 80 °C until measurement.

### 2.5. Telomere Length

Telomere length was measured as described by Ennour-Idrissi et al. [62], with some modifications. DNA was extracted from leucocytes using the Gentra PureGene Cell Kit (QIAGEN Inc., Canada), according to the manufacturer’s protocol. DNA quality and quantity were assessed using the NanoDrop12000c spectrophotometer (Thermo Scientific, Fisher Scientific Canada). Mean relative telomere length was measured using a quantitative polymerase chain reaction (qPCR) method, first described by Cawthon [63], with slight modifications. In short, both the genetic material of interest (telomere) and the single copy gene human beta-globin (Hbg) were amplified in quadruplicates, and the mean value was calculated. A negative control (no DNA template), reference DNA sample for normalisation between the experiments, and two cell line samples (one low-passage for longer telomeres and one high-passage for shorter telomeres) were run in each batch. Standard curves for telomere and Hbg amplifications were done using this same reference DNA sample: the efficiency was 99% and 91%, respectively. The mean cycle threshold (Ct) values were calculated using the three closest values of quadruplicate samples, with exclusion of the fourth value when it fell outside two standard deviations (SD) from the mean [64]. The intra-assay coefficient of variation (CV) of the mean was 1.45% and 1.78%, and the inter-assay CV was 2.80% and 2.04% for telomere and Hbg, respectively. The relative T/S ratio was calculated using this formula: relative T/S ratio = 2 − (mean telomere − mean Hbg) of sample − (mean telomere − mean Hbg) of the reference DNA [65]. All assays were performed blinded to the participants’ characteristics and clinical data. The T/S ratio was logarithmically transformed (base 2) for interpretation purposes, such that an increase of one unit of log2 (ratio T/S) can be interpreted as a two-fold longer telomere.

### 2.6. Inflammatory Biomarkers

Serum CRP concentrations were obtained by a highly sensitive immunoturbidimetric assay on an automated Modular Roche Diagnostics platform (System ExP Modular Hitachi). The coefficient of variation (CV) for CRP was <5%. Serum concentrations of IL-6 were obtained by ELISA (enzyme-linked immunosorbent assay), according to the maker’s protocol (R&D Systems, Minneapolis, USA), and the fluorescent signal was measured with an automated fluorimeter (Fluoros-Kan, LabSystems). Each sample was analysed in duplicate, and the inter- and intra-batch CVs were 12.7% and 5.3%, respectively. CRP and IL-6 concentrations were log-transformed, because their distributions were right skewed. An inflammatory index was created by summing standardised log-concentrations of CRP and IL-6 [44]. A positive score indicated higher inflammation.

### 2.7. Covariates

Based on the literature, covariates that might confound or modify the evaluated associations, were identified. Covariates collected at T1 were selected to respect the temporal sequence of events and minimize overadjustment, except for formal education, for which T3 data were used, as this variable was assessed in the numbers of years of schooling during a face-to-face interview. Age, sex, smoking status (never smoked regularly, former regular, current occasional or current regular smokers), alcohol intake (number of drinks per week), physical activity (frequency of 30 min leisure-time physical activity per week), diabetes, cardiovascular diseases (heart diseases, stroke, angina), and social support outside of work (number of confidants, number of helpers, relationship with spouse, and relationship with children) were evaluated using a self-report questionnaire [57]. Height, weight, hip and waist circumferences were measured in-person, using standardised protocols. Body mass index (BMI) (kg/m^2^) and waist-to-hip ratio were calculated and used as continuous variables. Blood pressure was measured following recognised guidelines [66,67]. Hypertension was defined by either systolic blood pressure >140 mmHg, diastolic blood pressure >90 mmHg [66,67], report of diagnosed hypertension, or use of antihypertensive medication.

### 2.8. Statistical Analysis

Associations between each work-related psychosocial factor and telomere length or the inflammatory index, and between telomere length or the inflammatory index and cognitive performance were evaluated using linear regression models with a robust variance estimator, using the generalised estimating equations (GEE) method. The potential effect modification of the inflammatory index in the relationship between telomere length and cognitive performance was tested with an interaction term. The exploration of a potential quadratic interaction in this relationship was tested by adding the square of the inflammatory index and an interaction term between the square of the inflammatory index and telomere length. All models were fitted globally and stratified by sex, while adjusting for the covariates previously described. The same procedure was used to estimate the total effect of each work-related psychosocial factor with global cognitive function, i.e., the total effect of these factors on cognitive performance, including direct and indirect effects through any potential mediator.

The Vansteelandt and Daniel method for multiple mediators was used to estimate direct, indirect and residual indirect effects [68]. This method accounts for multiple mediators, even when they are correlated with each other and when the structure of this correlation is unknown. The direct effect represents the association between each work-related psychosocial factor and cognitive performance that is not mediated by the telomere length or the inflammatory index. The indirect effect represents an association between each work-related factor and cognitive performance that is mediated by the telomere length or the inflammatory index, each considered separately. The residual indirect effect represents the association between each work-related factor and cognitive performance that is mediated through the interrelation between the telomere length and the inflammatory index, no matter the structure of this interrelation. This method was chosen given a priori evidence of a strong interrelation between the two mediators [11] that is generally a violation of the default assumptions for standard mediation analysis methods [68]. Standard errors for these estimated direct and indirect effects were obtained by bootstrap with 199 replications.

Multiple imputation (MI), using chained equations [69], was conducted for: (1) missing data in T1 variables among 3618 participants (3411 participants randomly selected for the study of biomarkers, plus 207 participants not included in the random selection); (2) missing data in T2 variables among the 3180 participants that were selected for the study of biomarkers and were either present in T2 data collection (n = 3087), or not present in T2, but present in the in-person data collection at T3 with cognitive data (n = 93); and (3) missing data in T3 variables among the 2219 participants included in the present study. Sixty imputations were computed, because 60% of the study population had at least one missing datum on a variable [69]. Inverse probability of censoring weights (IPCWs) were computed to correct for the differences in the characteristics between included participants and those lost to follow-up between each measurement time. IPCWs were calculated using predicted values obtained from logistic regressions of the probability of being censored between T1 and T2, and between T2 and T3, according to exposure and specific covariates at T1, and at T1 and T2, respectively [70]. Weights were recomputed for each bootstrap sample. Results from each imputed dataset were combined, producing average estimated effects and Wald-type confidence intervals. For more detailed information about the mediation analysis, MI and IPCW methods, their inclusion in the mediation analysis and the predictors used in each model, see Appendix A for statistical analyses.

The procedure to estimate direct and indirect effects was implemented in R version 4.1.3. All other analyses were performed using the SAS software, version 9.4.

## 3. Results

### 3.1. Characteristics of the Study Population

Among the 8981 eligible participants for follow-up (Figure 1), those randomly selected for the study of biomarkers (n = 3411) were similar to those not selected (n = 5570) (Appendix A). Among the participants randomly selected (n = 3411), those included in the present analysis (n = 2219; 51.1% females) were more likely to have a university degree and were less likely to be office workers, regular smokers, or to present with a diagnosis of diabetes or hypertension compared to those not included (n = 1192 (959 + 233 deceased), Figure 1 and Appendix A).

The mean follow-up time from T2 to T3 was 16.8 (standard deviation: 1.4) years and the mean age of participants at T2 (baseline for exposure) was 46.5 (7.9) years (Table 1). Compared to males, females were slightly younger, less educated and more often exposed to passive work, high job strain, low job control and iso-strain. The exposures to high psychological demands, low social support at work and ERI were similar for males and females. At T3, the mean values for education and the MoCA score were 15.3 (2.9) years completed and 25.6 (2.6), respectively.

### 3.2. Longitudinal Associations of Work-Related Psychosocial Factors with Telomere Length and Inflammatory Index

Associations were observed between exposure to passive work or low job control and shorter telomeres (β = −0.04, 95% CI −0.08 to −0.00 and β = −0.04, −0.07 to 0.00, respectively, Table 2). These same associations were stronger in females (β = −0.08, 95% CI −0.13 to −0.03 and β = −0.08, −0.13 to −0.02, respectively). No statistically significant associations between other work-related psychosocial factors and telomere length were observed overall or in females, and no statistically significant associations were observed in males. Regarding the inflammatory index, statistically significant associations were observed between low social support at work, ERI or iso-strain, and a higher inflammatory index among males only (β = 0.17, 95% CI 0.03 to 0.31, β = 0.14, 0.00 to 0.29, and β = 0.32, 0.10 to 0.55, respectively). No statistically significant associations with other work-related psychosocial factors and the inflammatory index were observed in males, and no statistically significant associations were observed overall or in females.

### 3.3. Cross-Sectional (T3) Associations of Telomere Length and Inflammatory Index with Cognitive Function

An association was observed between longer telomeres and higher cognitive performance (β = 0.26, 95% CI −0.00 to 0.52, *p* = 0.0527, Table 3). This association was stronger in males (β = 0.50, 95% CI 0.15 to 0.86). No statistically significant associations were observed between the inflammatory index and cognitive function, overall or by sex. Exploration of the effect modification of the inflammatory index on the association between telomeres and cognitive function showed a quadratic interaction and a U-shaped modifying association (Appendix A). The association between telomere length and cognitive performance was positive for participants with a low and high inflammatory index, but null or negative for middle values of the inflammatory index.

### 3.4. Mediating Effects of Telomere Length and Inflammatory Index

Overall, total and direct associations were observed between passive work and a lower cognitive performance (β_total_ = −0.30, 95% CI −0.55 to −0.06, and β_direct_ = −0.29, −0.54 to −0.04, Table 4), and between high psychological demands and higher cognitive performance (β_total_ = 0.23, 95% CI 0.03 to 0.44, and β_direct_ = 0.23, 0.02 to 0.43). In stratified analyses, these associations were driven by male participants (β_total_ = −0.40, 95% CI −0.78 to −0.02, and β_direct_ = −0.40, 0.78 to −0.03 for passive work; β_total_ = 0.38, 95% CI 0.09 to 0.67, and β_direct_ = 0.39, 0.09 to 0.68 for high demands). Total and direct associations were also observed between high job strain and higher cognitive performance in females (β_total and direct_ = 0.34, 95% CI 0.01 to 0.68), and between low reward and lower cognitive performance in males (β_total_ = −0.35, 95% CI −0.69 to −0.01 and β_direct_ = −0.32, −0.67 to 0.03). None of these associations were mediated by telomere length or the inflammatory index. No statistically significant total effect or direct effect were observed with low job control, low social support at work, ERI, or iso-strain. No statistically significant indirect or residual effects were detected overall, or by sex. Figure 2 illustrates an example of the total, direct, indirect, and residual effects through telomere length and inflammatory index, between passive work and cognitive function.

## 4. Discussion

### 4.1. Interpretation of Results

In this 17-year longitudinal study of more than 2000 white-collar workers, it was found that passive work in the study sample and low reward in males were associated with a poorer cognitive performance, whereas high psychological demands in the study sample and high job strain in females were associated with a better cognitive performance. However, none of these associations were mediated by telomere length or the inflammatory index. To our knowledge, this is the first study evaluating the indirect effects of work-related psychosocial factors on global cognitive function through biological mechanisms, including telomere length and low-grade inflammation. This is also the first longitudinal study evaluating the association between work-related psychosocial factors and telomere length.

Only two studies [41,71] had previously evaluated the possible mediating effects of inflammatory biomarkers on the relationship between work-related psychosocial factors and other health outcomes associated with cognitive function [2,72]. The first study was conducted among a subsample of 2101 white-collar workers from the Whitehall II study, aged 50 years on average at the time of exposure measurement, and followed for 10 years [41]. This study reported a weak indirect effect for the association between low social support at work and diabetes among females though IL-6 plasma concentration. The second was a cross-sectional study conducted among a small sample of 204 young male adult workers from veterinary, agricultural, textile or poultry industries in Jordan [71]. This study found that the association between exposure to ERI and the metabolic syndrome was mediated by CRP serum concentrations.

The fact that we found no indirect effects through these biomarkers suggests that other biological pathways may be involved in the association between work-related psychosocial factors and cognitive function. The associations found with exposure to passive work and poorer cognitive performance, or with exposure to high psychological demand and better cognitive performance, are aligned with the cognitive reserve theory [73]. A higher level of education, as well as activities with greater intellectual complexity, or employment that requires more complex skills, may promote the development and the efficiency of the brain, enhancing cognitive reserve and protecting against premature cognitive decline [74]. Passive work is characterized by work with a low cognitive stimulation, in terms of low psychological demands, repetitive work, and low skill utilization and development (i.e., low job control), and is thus less favorable for the preservation of cognitive reserve. A systematic review with meta-analysis reported that exposure to high complexity at work was associated with a lower risk of dementia compared to low complexity at work, exposure to passive work or job strain was associated with a faster cognitive decline compared to active work, and that exposure to high psychological demands was associated with a slower cognitive decline compared to low psychological demands [6]. Moreover, longitudinal studies that have evaluated the association between exposure to passive or active work and cognitive function or dementia, found similar associations as those observed in the present study. In these studies, associations were observed with exposure to passive work and lower global cognitive function [36], lower performance on neuropsychological tests [75,76], or higher risk of dementia [77,78]. Two longitudinal studies also found protective associations between active work and global cognitive function [37] or dementia [79]. Negative findings were reported in two longitudinal studies for the association between passive work and global cognitive function [39] or annual decline of neuropsychological scores [80]. For the most part, these results support the cognitive reserve theory as a plausible explanation for the effect of work-related psychosocial factors on long-term cognitive function.

Although mediation effects were not supported in the current study, associations were observed between specific work-related psychosocial factors and shorter telomeres, especially among females, along with a higher inflammatory index, especially among males, suggesting that these factors can increase systemic low-grade inflammation and oxidation. Only two previous cross-sectional studies have examined the association between these factors and telomere length and their results are difficult to compare to ours, because neither examined potential sex differences [81,82]. The first study was conducted among a sample of 435 workers, aged 61 years on average, from the Multiethnic Study of Atherosclerosis (MESA) study [81]. No association was found between exposure to high job strain, active work, passive work, high psychological demands or low job control evaluated (with a validated questionnaire and a job exposure matrix) and telomere length measured with the T/S ratio. The second study was conducted among a sample of 141 employees working in geriatric care units in Germany, aged 44 years on average [82]. No association was found between exposure to high psychological demands, low job control or low social support at work (evaluated with a validated questionnaire) and telomere length (expressed as relative length compared to albumin DNA). In our study, no association was found between high job strain, high psychological demands, and low social support at work, and telomere length. However, exposure to passive work or low job control was associated with shorter telomeres, especially among females.

The current findings suggest that some associations between exposures and outcomes varies according to biological sex. Females tend to be more exposed to psychosocial stressors at work [45] and have longer telomeres [47]. Moreover, sex differences have been suggested in the effect of chronic stress on the inflammatory and oxidative response to these exposures [46]. In another sample from the MESA study conducted among 1029 participants, sex differences were observed in the association between chronic stress exposure and telomere shortening, using the T/S ratio [83]. Similar to our findings here, the association between greater chronic stress and telomere shortening was stronger among females than males [83]. These results suggested that some differences between males and females could be present, but the body of knowledge is still insufficient to draw solid conclusions.

This study is the first to observe a U-shaped modifying effect of inflammatory biomarkers in the association between telomere length and global cognitive function. One previous longitudinal study conducted among a sample of 497 young adult residents of Jerusalem evaluated the modifying effect of an inflammatory index, combining CRP, fibrinogen and a number of leukocytes on the association between telomere shortening over 5 to 10 years, and global cognitive function evaluated with a neurocognitive test battery [84]. The authors reported an association between faster telomere shortening and poorer cognitive function, but the association was not mediated by inflammation. Nevertheless, considering that telomere shortening and low-grade inflammation are biologically interrelated [11], this effect modification needs to be explored in future studies.

### 4.2. Strengths and Limitations

This study has several strengths, including a large sample size, with similar number of males and females for the evaluation of sex differences, and a longitudinal design. Work-related psychosocial factor exposure was measured with validated instruments and well before the assessment of cognitive function and biomarkers. Rigorous and innovative statistical methods were employed to evaluate the indirect effects of telomeres and inflammation in the presence of correlation and interaction between these mediators, and to correct for the potential selection bias. Moreover, several potential confounders were measured before exposure assessment and were controlled for in statistical models, which minimized the potential of overadjustment by mediators.

This study also has limitations. First, telomere length, inflammatory biomarkers and global cognitive function were all measured at the end of follow-up. Thus, causal inference about the associations observed between these variables cannot be made. Variation in telomere length and in inflammatory biomarkers can either be a cause or a consequence of the variation in global cognitive function. However, as several previous longitudinal studies have observed associations between inflammatory biomarkers or telomere length and cognitive function [13,14,15,16,17,18,19,20] or dementia [21,22,23,24], a causal hypothesis is supported by a priori evidence. Second, cognitive function was not evaluated at the beginning of the study, thus it was not possible to control for cognitive performance at baseline and change in cognition over time could not be determined. However, the fact that participants were relatively young, with only 4% aged 65 years or older, and that 90% were still working at T2, does not favor the presence of cognitive impairment at T2, although it cannot be completely ruled out. Third, participants may have changed jobs and working conditions during their career over the 17-year follow-up. We do not have this information and misclassification of the exposure over time is possible. However, due to the good working conditions generally offered to white-collar workers, our study population tended to maintain their occupational position over time [44]. Nevertheless, the possibility that exposed participants may have been more likely to quit or change their job cannot be excluded. This would result in a potential selection bias, that generally tends to underestimate the associations. Fourth, the healthy worker effect could be present, as healthier workers and those less exposed to work-related psychosocial factors tend to stay longer in the labour force [85,86], and if so, could lead to an underestimation of the observed associations [85,86,87]. While statistical analyses partially controlled for the healthy worker effect during the course of the follow-up, using IPCW for death and loss to follow-up, underestimation remains possible [87]. Fifth, multiple testing may have increased the risk of false discovery, although all comparisons were determined a priori and were scientifically justified. Finally, the study population was composed of white-collar workers and mostly Caucasians, and thus may not be representative of the entire workforce population. Caution should be exercised in generalizing results. However, participants held a wide variety of jobs, such as office workers, technicians, professionals, and managers, ensuring exposure diversity, while controlling for the physicochemical exposures present in blue-collar workers. White-collar workers constitute the largest segment of the Canadian workforce, a fact which contributes to increasing the scope of the results.

## 5. Conclusions

This study suggests that some work-related psychosocial factors could be associated with shorter telomeres and low-grade inflammation, but these associations do not explain the relationship between work-related psychosocial factors and global cognitive function. Further research on the biological pathways by which these factors affect cognitive function could help identify which specific work-related psychosocial factors with positive effects could be targeted for enhancement, with the potential of increasing cognitive reserve. Similarly, factors with negative effects could be targeted for reduction, in order to prevent systemic low-grade inflammation, oxidation, and cellular aging. Interventions already exist to reduce occupational exposures. Work-related psychosocial factors, along with other modifiable lifestyle risk factors, including physical activity, diet, and vascular health, could be part of future prevention strategies aimed at maintaining cognitive function and promoting healthy aging.

## Figures and Tables

**Figure 1 ijerph-20-04929-f001:**
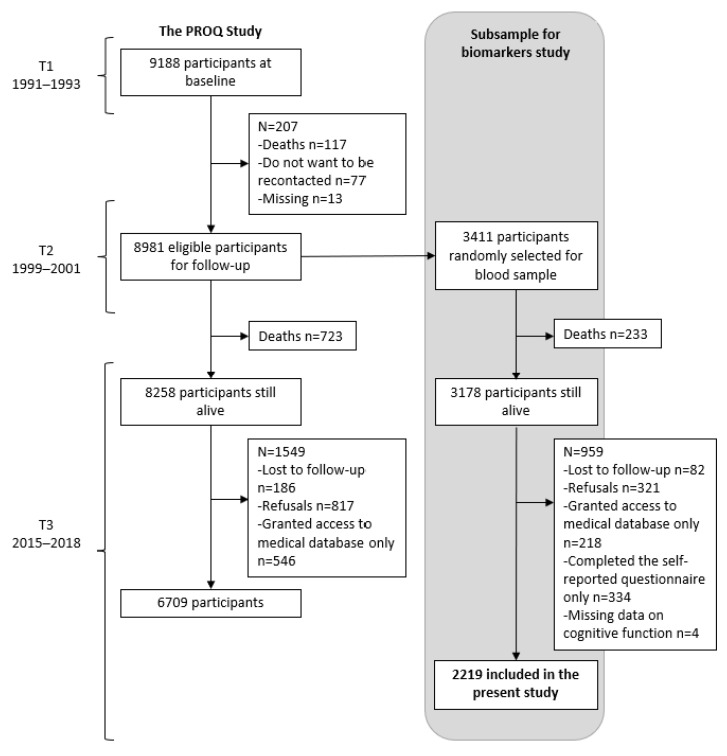
Flow chart of the study population.

**Figure 2 ijerph-20-04929-f002:**
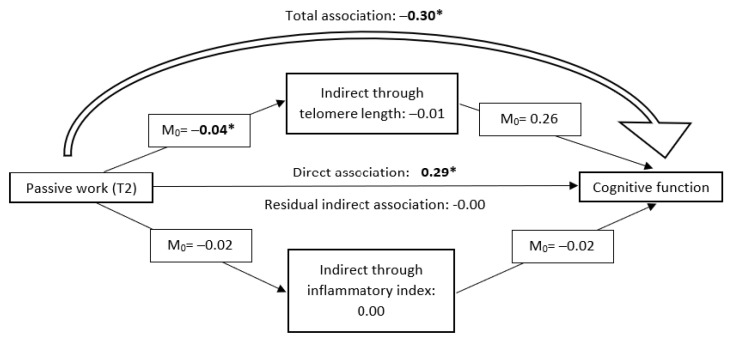
Illustration of the total, direct, and indirect associations through telomere length and inflammatory index, between exposure to passive work and global cognitive function. M_0_: Associations between exposure and mediators, or between mediators and global cognitive function. All associations were adjusted for age, sex, education, BMI, smoking, alcohol, physical activity, diabetes, hypertension, cardiovascular diseases, number of confidants, number of helpers, relationship with spouse, and relationship with children. Direct associations were additionally adjusted for inflammatory index, telomere length, inflammatory index squared, and interaction terms between telomere length and inflammatory index, and between telomere length and inflammatory index squared. Missing data on covariates, exposure, and outcomes were imputed with multiple imputation, and data were weighted for lost to follow-up between T1 and T2, and between T2 and T3, with an inverse probability of censoring weighting. * *p* < 0.05.

**Table 1 ijerph-20-04929-t001:** Baseline characteristics ^a^ of the study sample ^b^, overall and by sex.

	Overall ^c^ N = 2219 Mean ± SD or n (%)	Females N = 1133 (51%) Mean ± SD or n (%)	Males N = 1086 (49%) Mean ± SD or n (%)
Age (y)			
At T1 (1991–1993)	38.9 ± 7.9	37.4 ± 7.4	40.4 ± 8.0
At T2 (1999–2001)	46.5 ± 7.9	45.0 ± 7.4	48.0 ± 8.1
At T3 (2015–2018)	63.2 ± 7.4	61.8 ± 7.0	64.7 ± 7.6
Education at T3 (y)	15.3 ± 2.9	14.3 ± 2.7	16.3 ± 2.7
Smoking status			
Never smoked regularly	1012 (46.0)	546 (48.5)	466 (43.4)
Former regular smoker	735 (33.4)	330 (29.3)	405 (37.7)
Current occasional smoker ^d^	101 (4.6)	55 (4.9)	46 (4.3)
Current regular smoker ^d^	351 (16.0)	195 (17.3)	156 (14.5)
Alcohol intake (drinks/week)	3.5 ± 4.5	2.3 ± 2.9	4.7 ± 5.4
Physical activity (times/month)	5.3 ± 4.3	4.7 ± 4.2	6.0 ± 4.4
Body mass index (kg/m^2^)	24.5 ± 3.8	23.5 ± 3.9	25.5 ± 3.5
Waist-to-hip ratio	0.84 ± 0.1	0.76 ± 0.07	0.92 ± 0.05
Diabetes (yes)	34 (1.5)	22 (1.9)	12 (1.1)
Hypertension (yes)	347 (15.6)	120 (10.6)	227 (20.9)
Cardiovascular diseases (yes)	52 (2.3)	26 (2.3)	26 (2.4)
Confidants (number)	2.1 ± 1.5	2.3 ± 1.4	1.9 ± 1.5
Helpers (number)	3.0 ± 1.6	3.1 ± 1.5	2.9 ± 1.7
Relationship with spouse			
Satisfying	1370 (64.8)	647 (59.6)	723 (70.3)
Unsatisfying	215 (10.2)	112 (10.3)	103 (10.0)
No spouse	528 (25.0)	326 (30.1)	202 (19.7)
Relationship with children			
Satisfying	1311 (61.1)	611 (55.4)	310 (29.7)
Unsatisfying	64 (3.0)	30 (2.7)	700 (67.1)
No children	771 (35.9)	461 (41.8)	34 (3.3)
Passive work at T2	722 (34.4)	423 (40.0)	299 (28.7)
High job strain at T2	396 (18.9)	239 (22.6)	157 (15.1)
High psychological demands at T2	991 (46.9)	482 (45.2)	509 (48.5)
Low job control at T2	1120 (53.3)	664 (62.7)	456 (43.7)
Low social support at work at T2	1152 (55.2)	584 (55.6)	568 (54.8)
Effort–reward imbalance at T2	497 (24.3)	255 (24.7)	242 (23.8)
Low reward at T2	954 (46.6)	491 (47.5)	463 (45.6)
Iso-strain at T2	275 (13.3)	156 (15.0)	119 (11.5)
CRP concentration at T3, mg/L, median, interquartile range	1.15 (0.43;2.65)	1.34 (0.48;3.20)	0.99 (0.41;2.15)
IL-6 concentration at T3, pg/mL, median, interquartile range	2.05 (1.38;3.20)	2.01 (1.35;3.18)	2.09 (1.41;3.2)
Telomere length at T3, ratio T/S median, interquartile range	0.75 (0.63;0.89)	0.76 (0.64;0.92)	0.74 (0.61;0.86)
Inflammatory index, mean, min, max	0.00 (−4.87;5.22)	0.03 (−4.87;5.22)	−0.03 (−2.95;5.02)
MoCA score (out of 30) at T3	25.6 ± 2.6	25.8 ± 2.6	25.4 ± 2.5

^a^ Measured at T1, unless otherwise specified. ^b^ Participants selected for blood sample, who participated at T3 and had cognitive data. ^c^ Less than 5% of missing data on any variable, except for work-related psychosocial factors (6–8%). ^d^ Occasional smoker refers to reporting smoking, but not everyday; regular smoker refers to reporting smoking everyday.

**Table 2 ijerph-20-04929-t002:** Longitudinal associations of work-related psychosocial factors (T2) with telomere length and inflammatory index (T3), overall and by sex.

	Telomere Length ^a^ β (95% CI) *p*-Value		
	Overall	Females	Males
Passive work	**−0.04 (−0.08;−0.00) *p* = 0.0263**	**−0.08 (−0.13;−0.03) *p* = 0.0026**	0.01 (−0.05;0.06) *p* = 0.8237
High job strain	0.01 (−0.04;0.05) *p* = 0.7839	0.01 (−0.05;0.07) *p* = 0.7218	0.00 (−0.07;0.07) *p* = 0.9921
High psychological demand	0.02 (−0.02;0.05) *p* = 0.3770	0.04 (−0.01;0.09) *p* = 0.1103	−0.01 (−0.06;0.04) *p* = 0.6403
Low job control	−0.04 (−0.07;0.00) *p* = 0.0533	**−0.08 (−0.13;−0.02) *p* = 0.0051**	0.01 (−0.04;0.05) *p* = 0.8351
Low social support	−0.01 (−0.04;0.03 *p* = 0.7626)	−0.01 (−0.06;0.04) *p* = 0.7497	−0.00 (−0.05;0.04) *p* = 0.8895
Effort–reward imbalance	0.02 (−0.01;0.06) *p* = 0.2232	0.03 (−0.02;0.08) *p* = 0.2987	0.01 (−0.04;0.06) *p* = 0.6262
Low reward	−0.03 (−0.06;0.01) *p* = 0.1166	−0.04 (−0.09;0.01) *p* = 0.1021	−0.02 (−0.07;0.03) *p* = 0.4544
Iso-strain	0.00 (−0.05;0.05) *p* = 0.9624	−0.07 (−0.07;0.06) *p* = 0.9325	0.00 (−0.08;0.08) *p* = 0.9393
	**Inflammatory index ^a^** **β (95% CI) *p*-value**		
Passive work	−0.01 (−0.11;0.10) *p* = 0.8809	0.04 (−0.11;0.19) *p* = 0.6134	−0.07 (−0.22;0.08) *p* = 0.3633
High job strain	0.07 (−0.06;0.20) *p* = 0.2588	0.02 (−0.15;0.19) *p* = 0.7898	0.18 (−0.02;0.38) *p* = 0.0759
High psychological demand	−0.01 (−0.11;0.09) *p* = 0.9126	−0.03 (−0.18;0.11) *p* = 0.6610	0.04 (−0.09;0.18) *p* = 0.5506
Low job control	0.04 (−0.06;0.14) *p* = 0.4275	0.06 (−0.09;0.21) *p* = 0.4380	0.04 (−0.10;0.17) *p* = 0.6049
Low social support	0.07 (−0.03;0.017) *p* = 0.1579	−0.01 (−0.15;0.14) *p* = 0.9263	**0.17 (0.03;0.31) *p* = 0.0142**
Effort–reward imbalance	0.06 (−0.04;0.16) *p* = 0.2482	0.00 (−0.15;0.15) *p* = 0.9939	**0.14 (0.00;0.29) *p* = 0.0487**
Low reward	0.07 (−0.03;0.17) *p* = 0.1962	0.02 (−0.12;0.16) *p* = 0.7419	0.13 (−0.01;0.27) *p* = 0.0603
Iso-strain	0.10 (−0.05;0.25) *p* = 0.1749	−0.03 (−0.23;0.17) *p* = 0.7435	**0.32 (0.10;0.55) *p* = 0.0038**

^a^ Adjusted for sex (except for stratified analyses), age, years of education, BMI, smoking, alcohol, physical activity, diabetes, hypertension, cardiovascular diseases, number of confidants, number of helpers, relationship with spouse, and relationship with children. Missing data on covariates, exposure, and outcomes were imputed with multiple imputation, and data were weighted for loss to follow-up between T1 and T2, and between T2 and T3, with an inverse probability of censoring weighting. β = beta coefficient, CI = confidence interval, BMI = body mass index.

**Table 3 ijerph-20-04929-t003:** Cross-sectional associations of telomere length and inflammatory index with cognitive function at T3, overall and by sex.

	Overall β (95% CI) *p*-Value	Females β (95% CI) *p*-Value	Males β (95% CI) *p*-Value
Telomere length ^a^	0.26 (-0.00;0.52) *p* = 0.0527	0.10 (−0.27;0.47) *p* = 0.5923	**0.50 (0.15;0.86) *p* = 0.0058**
Inflammatory index ^a^	−0.02 (-0.11;0.08) *p* = 0.7425	0.07 (−0.06;0.20) *p* = 0.2740	−0.12 (−0.25;0.01) *p* = 0.0763

^a^ Adjusted for sex (except for stratified analyses), age, years of education, BMI, smoking, alcohol, physical activity, diabetes, hypertension, cardiovascular diseases, number of confidants, number of helpers, relationship with spouse, and relationship with children. When including interaction terms between telomere length and inflammatory index in quartiles, the *p*-values for the interaction between telomere length and inflammatory index were overall = 0.0283, females = 0.0050 and males = 0.2003. Missing data on covariates, exposure, and outcomes were imputed with multiple imputation, and data were weighted for loss to follow-up between T1 and T2, and between T2 and T3, with an inverse probability of censoring weighting. β = beta coefficient, CI = confidence interval, BMI = body mass index.

**Table 4 ijerph-20-04929-t004:** Association between work-related psychosocial factors at T2 and global cognitive function at T3: total, direct, and indirect ^a^ associations through telomere length and inflammatory index, overall and by sex.

OVERALL	Total ^b^ β (95% CI)	Direct ^c^ β (95% CI)	Indirect Telomere length ^b^ β (95% CI)	Indirect Inflammatory index ^b^ β (95% CI)
Passive work	**−0.30 (−0.55;−0.06) *p* = 0.0163**	**−0.29 (−0.54;−0.04) *p* = 0.0208**	−0.01 (−0.03;0.01) *p* = 0.2697	−0.00 (−0.01;0.01) *p* = 0.9928
High job strain	0.24 (−0.03;0.50) *p* = 0.0764	0.23 (−0.04;0.49) *p* = 0.0911	0.00 (−0.01;0.02) *p* = 0.8171	0.00 (−0.01;0.02) *p* = 0.9236
High psychological demand	**0.23 (0.03;0.44) *p* = 0.0280**	**0.23 (0.02;0.43) *p* = 0.0348**	0.00 (−0.01;0.02) *p* = 0.5107	−0.00 (−0.01;0.01) *p* = 0.9837
Low job control	−0.13 (−0.36;0.10) *p* = 0.2571	−0.13 (−0.36;0.10) *p* = 0.2726	−0.01 (−0.02;0.01) *p* = 0.2422	0.00 (−0.01;0.02) *p* = 0.9497
Low social support	−0.15 (−0.37;0.08) *p* = 0.2051	−0.16 (−0.38;0.07) *p* = 0.1739	−0.00 (−0.01;0.01) *p* = 0.7874	0.00 (−0.01;0.01) *p* = 0.8831
Effort–reward imbalance	0.12 (−0.10;0.34) *p* = 0.2904	0.11 (−0.11;0.34) *p* = 0.3247	0.01 (−0.01;0.02) *p* = 0.3969	0.00 (−0.01;0.01) *p* = 0.9483
Low reward	−0.21 (−0.44;0.02) *p* = 0.0708	−0.20 (−0.43;0.03) *p* = 0.0828	−0.01 (−0.02;0.01) *p* = 0.2945	0.00 (−0.01;0.01) *p* = 0.8886
Iso-strain	0.19 (−0.12;0.51) *p* = 0.2343	0.19 (−0.13;0.50) *p* = 0.2529	0.00 (−0.01;0.02) *p* = 0.9705	0.00 (−0.02;0.02) *p* = 0.8834
**FEMALES**				
Passive work	−0.22 (−0.54;0.10) *p* = 0.1833	−0.21 (−0.54;0.11) *p* = 0.1943	−0.01 (−0.06;0.04) *p* = 0.6926	0.00 (−0.02;0.03) *p* = 0.8117
High job strain	**0.34 (0.01;0.68) *p* = 0.0449**	**0.34 (0.01;0.68) *p* = 0.0456**	0.00 (−0.02;0.02) *p* = 0.8929	0.00 (−0.03;0.03) *p* = 0.8929
High psychological demand	0.11 (−0.19;0.40) *p* = 0.4763	0.10 (−0.20;0.39) *p* = 0.5100	0.01 (−0.02;0.04) *p* = 0.6855	−0.00 (−0.03;0.02) *p* = 0.8380
Low job control	0.04 (−0.30;0.36) *p* = 0.8259	0.04 (−0.29;0.37) *p* = 0.8079	−0.01 (−0.05;0.03) *p* = 0.5501	0.00 (−0.02;0.03) *p* = 0.7316
Low social support	−0.02 (−0.33;0.29) *p* = 0.8968	−0.03 (−0.35;0.28) *p* = 0.8378	−0.00 (−0.02;0.01) *p* = 0.8465	−0.00 (−0.03;0.03) *p* = 0.9883
Effort–reward imbalance	0.05 (−0.25;0.35) *p* = 0.7535	0.04 (−0.26;0.35) *p* = 0.7879	0.00 (−0.02;0.02) *p* = 0.6873	0.00 (−0.02;0.02) *p* = 0.9920
Low reward	−0.09 (−0.39;0.21) *p* = 0.5472	−0.10 (−0.40;0.21) *p* = 0.5367	−0.01 (−0.16;0.14) *p* = 0.9303	0.00 (−0.01;0.01) *p* = 0.9788
Iso-strain	0.30 (−0.10;0.71) *p* = 0.1399	0.31 (−0.10;0.72) *p* = 0.1395	−0.00 (−0.02;0.02) *p* = 0.9927	−0.00 (−0.15;0.14) *p* = 0.9927
**MALES**				
Passive work	**−0.40 (−0.78;−0.02) *p* = 0.0368**	**−0.40 (−0.78;−0.03) *p* = 0.0366**	0.00 (−0.02;0.03) *p* = 0.8128	0.01 (−0.02;0.03) *p* = 0.5592
High job strain	0.11 (−0.29;0.51) *p* = 0.5883	0.12 (−0.29;0.52) *p* = 0.5630	0.00 (−0.03;0.03) *p* = 0.9985	−0.02 (−0.06;0.03) *p* = 0.4314
High psychological demand	**0.38 (0.09;0.67) *p* = 0.0104**	**0.39 (0.09;0.68) *p* = 0.0095**	−0.00 (−0.03;0.02) *p* = 0.7092	−0.00 (−0.02;0.01) *p* = 0.6562
Low job control	−0.28 (−0.61;0.05) *p* = 0.0981	−0.27 (−0.60;0.05) *p* = 0.1020	0.00 (−0.02;0.03) *p* = 0.8299	−0.00 (−0.02;0.01) *p* = 0.7254
Low social support	−0.23 (−0.55;0.09) *p* = 0.1661	−0.22 (−0.55;0.10) *p* = 0.9560	0.00 (−0.02;0.02) *p* = 0.9560	−0.01 (−0.05;0.02) *p* = 0.4338
Effort–reward imbalance	0.17 (−0.14;0.49) *p* = 0.2773	0.18 (−0.13;0.50) *p* = 0.2561	0.01 (−0.02;0.03) *p* = 0.6217	−0.01 (−0.04;0.02) *p* = 0.3853
Low reward	**−0.35 (−0.69;−0.01) *p* = 0.0459**	−0.32 (−0.67;0.03) *p* = 0.0702	−0.01 (−0.03;0.02) *p* = 0.6116	−0.01 (−0.04;0.02) *p* = 0.4543
Iso-strain	0.06 (−0.42;0.53) *p* = 0.8142	0.08 (−0.41;0.57) *p* = 0.7491	0.00 (−0.04;0.04) *p* = 0.8857	−0.03 (−0.10;0.04) *p* = 0.3536

^a^ All residual indirect associations showed β = 0.00 or −0.00 with 95% CI or −0.01 to 0.01 at most, and *p*-values from 0.7791 to 0.9978. ^b^ Adjusted for sex (except for stratified analyses), age, education, BMI, smoking, alcohol, physical activity, diabetes, hypertension, cardiovascular disease, number of confidants, number of helpers, relationship with spouse, and relationship with children. ^c^ Additionally adjusted for inflammatory index, telomere length, inflammatory index squared, and interaction terms between telomere length and inflammatory index, and between telomere length and inflammatory index squared. Missing data on covariates, exposures, and outcomes were imputed with multiple imputation, and data were weighted for lost to follow-up between T1 and T2, and between T2 and T3, with an inverse probability of censoring weighting. Β = beta coefficient, CI = confidence interval, BMI = body mass index.

## Data Availability

The data underlying this article cannot be shared publicly because of privacy concerns relative to the individuals who participated in the study. However, data will be shared following a justified request to the corresponding author, conditional on permission from the CHU de Québec-Université Laval Research Ethics Board.

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
