# Peer review of "Work-Related Psychosocial Factors and Global Cognitive Function: Are Telomere Length and Low-Grade Inflammation Potential Mediators of This Association?"

_ijerph, 2023, doi:10.3390/ijerph20064929_

Round 1

Reviewer 1 Report

The authors present a longitudinal study established over 24 years with an intermediate experimentation time after 8 years. This study includes a very large number of subjects (9188 subjects at the beginning) and many correlated parameters. All the questions I had during the reading were answered in the text below. So I have no major questions/comments. I think this study is worth considering.
I do have 2-3 minor comments:
- 2.1. Study design and data collection: how did you take into account, if at all, subjects who changed jobs and/or working conditions during their career? Also, over 24 years, working conditions have certainly changed. In general, the pressure on workers may have decreased/increased. Have you taken these 2 parameters into account?
- Table 2. To facilitate reading, p-values <0.05 could be put in bold.
- references: the layout of the references from 1 to 50 is not correct.

Author Response

Response to reviewer #1

The authors present a longitudinal study established over 24 years with an intermediate experimentation time after 8 years. This study includes a very large number of subjects (9188 subjects at the beginning) and many correlated parameters. All the questions I had during the reading were answered in the text below. So, I have no major questions/comments. I think this study is worth considering.
I do have 2-3 minor comments:

Point #1: - 2.1. Study design and data collection: how did you take into account, if at all, subjects who changed jobs and/or working conditions during their career? Also, over 24 years, working conditions have certainly changed. In general, the pressure on workers may have decreased/increased. Have you taken these 2 parameters into account?

Response to point #1:

Thank you for this comment. The fact that we do not have measured between T2 and T3 is a limitation of the present study. However, we did observe that between T1 and T2, the great majority (82%) of active participants kept the same occupation type and remained in the same organization. This is not surprising given that white-collar jobs usually offer good working conditions including income, medical insurance, and a retirement pension scheme1. This observation concerning job stability is reassuring in that it indicates that variations in these important basic working conditions over time were probably not that frequent. Given that most participants had retired by T3, it was not possible to measure changes in psychosocial work stressors between T2 and T3. However, we cannot exclude the possibility that exposed participants may have been more likely to quit or change their job, which would result in a potential selection bias that generally tends to underestimate the observed associations. We added this limitation in the manuscript.

“Third, participants may have changed jobs and working conditions during their career over the 17-year follow-up. We do not have this information and misclassification of the exposure over time is possible. However, due to the good working conditions generally offered to white-collar workers, our study population tended to maintain their occupational position over time.44 Nevertheless, the possibility that exposed participants may have been more likely to quit or change their job cannot be excluded. This would result in a potential selection bias that generally tends to underestimate the associations.”

  1. Duchaine CS, Brisson C, Talbot D, et al. Psychosocial stressors at work and inflammatory biomarkers: PROspective Quebec Study on Work and Health. 2021;133:105400.

Point #2: - Table 2. To facilitate reading, p-values <0.05 could be put in bold.

Response to point #2: The change has been made.

Point #3: - references: the layout of the references from 1 to 50 is not correct.

Response to point #3: The change has been made.

Reviewer 2 Report

This paper presents the measurement of some work-related psychosocial factors and their relationship with cognitive impairment in a very interesting way. Some aspects that require revision or clarification are the following:

Regarding the covariates, I would like to ask if the definition of "physical activity" as frequency of 30 minutes leisure-time physical activity per month is correct, or is it a typo, because it is a very low amount of time.

Loss of follow-up among the 3178 participants reached 30% (mainly due to: refusal, granted access to medical database only or completed the self-reported questionnaire only). This is a significant proportion that should be considered in the limitations of the study.

Tables 2 and 4 should be simplified, so that only results with statistical significance are shown.

Cognitive impairment is clearly multifactorial, and part of the explanation for the "discrete" study results of its relationship with some work-related psychosocial factors is due to this. Perhaps the need to combine these factors with genetic markers, eating habits, physical exercise or vaccinations and infectious pathology in the past should be introduced into the discussion.

Author Response

Response to reviewer #2

This paper presents the measurement of some work-related psychosocial factors and their relationship with cognitive impairment in a very interesting way. Some aspects that require revision or clarification are the following:

Point #1: Regarding the covariates, I would like to ask if the definition of "physical activity" as frequency of 30 minutes leisure-time physical activity per month is correct, or is it a typo, because it is a very low amount of time.

Response to point #1: Thank you very much to pointed out this typo. It was in fact 30 minutes per week. The change has been made.

Point #2: Loss of follow-up among the 3178 participants reached 30% (mainly due to: refusal, granted access to medical database only or completed the self-reported questionnaire only). This is a significant proportion that should be considered in the limitations of the study.

Response to point #2: Thank you for your comment. We previously acknowledged that loss to follow-up was a limitation that could result in a potential selection bias. As mentioned in our statistical analysis section, we mitigated this limitation by using inverse probability of censoring weight (IPCW): Inverse probability of censoring weights (IPCW) were computed to correct for the differences in the characteristics between included participants and those lost to follow-up between each measurement time. IPCWs were calculated using predicted values obtained from logistic regressions of the probability of being censored between T1 and T2, and between T2 and T3, according to exposure and specific covariates at T1, and at T1 and T2, respectively.”

 We agree that this method could not completely rule out the potential of selection bias due to loss to follow-up. As mentioned in our limitation section, this bias tend to result in an underestimation of the association as healthier workers and those less exposed to work-related psychosocial factors tend to stay longer in the labour force: Fourth, the healthy worker effect could be present, as healthier workers and those less exposed to work-related psychosocial factors tend to stay longer in the labour force, and if so, could lead to underestimation of the observed association. While statistical analyses partially controlled for the healthy worker effect during the course of follow-up using IPCW for death and loss to follow-up, underestimation remains possible.”

Point #3: Tables 2 and 4 should be simplified, so that only results with statistical significance are shown.

Response to point #3: We do not think that only presenting statistically significant results is the best practice for results reports in research. According to the recognized STROBE guidelines, all results should be reported: https://www.equator-network.org/reporting-guidelines/strobe/

Point #4: Cognitive impairment is clearly multifactorial, and part of the explanation for the "discrete" study results of its relationship with some work-related psychosocial factors is due to this. Perhaps the need to combine these factors with genetic markers, eating habits, physical exercise or vaccinations and infectious pathology in the past should be introduced into the discussion.

Response to point #4: We fully agree that cognitive impairment is multifactorial. In our study we focused on the effect of work-related psychosocial factors and tried to isolate the effect of this exposure by adjusting for several potential confounders. In fact, lifestyle habit (smoking status, alcohol intake, physical activity, and diet in sensitivity analyses) and comorbidities (diabetes, cardiovascular diseases (heart diseases, stroke, angina), hypertension) were controlled in the analyses. We are aware that work-related psychosocial factors constitute only one aspect in the global preventive strategies of neurocognitive disorders. As suggested by the reviewer, we added other modifiable factors that should be considered for the prevention of the cognitive function:

“Work-related psychosocial factors, along with other modifiable lifestyle risk factors including physical activity, diet, and vascular health, could be part of future prevention strategies aimed at maintaining cognitive function and promoting healthy aging.”